# Artificial intelligence as a diagnostic aid in cross-sectional radiological imaging of the abdominopelvic cavity: a protocol for a systematic review

George E Fowler [ORCID],[1] Rhiannon C Macefield [ORCID],[1] Conor Hardacre,[1] Mark P Callaway,[2] Neil J Smart,[3] Natalie S Blencowe [ORCID] [1]

¹Centre for Surgical Research, Population Health Sciences, Bristol Medical School, University of Bristol, Bristol, UK
²Department of Clinical Radiology, Bristol Royal Infirmary, Bristol, UK
³Exeter Surgical Health Services Research Unit (HeSRU), Royal Devon and Exeter NHS Foundation Trust, Exeter, UK

**Correspondence to**
Dr George E Fowler;
george.fowler@bristol.ac.uk

## ABSTRACT

**Introduction** The application of artificial intelligence (AI) technologies as a diagnostic aid in healthcare is increasing. Benefits include applications to improve health systems, such as rapid and accurate interpretation of medical images. This may improve the performance of diagnostic, prognostic and management decisions. While a large amount of work has been undertaken discussing the role of AI little is understood regarding the performance of such applications in the clinical setting. This systematic review aims to critically appraise the diagnostic performance of AI algorithms to identify disease from cross-sectional radiological images of the abdominopelvic cavity, to identify current limitations and inform future research.

**Methods and analysis** A systematic search will be conducted on Medline, EMBASE and the Cochrane Central Register of Controlled Trials to identify relevant studies. Primary studies where AI-based technologies have been used as a diagnostic aid in cross-sectional radiological images of the abdominopelvic cavity will be included. Diagnostic accuracy of AI models, including reported sensitivity, specificity, predictive values, likelihood ratios and the area under the receiver operating characteristic curve will be examined and compared with standard practice. Risk of bias of included studies will be assessed using the QUADAS-2 tool. Findings will be reported according to the Synthesis Without Meta-analysis guidelines.

**Ethics and dissemination** No ethical approval is required as primary data will not be collected. The results will inform further research studies in this field. Findings will be disseminated at relevant conferences, on social media and published in a peer-reviewed journal.

**PROSPERO registration number** CRD42021237249.

## Strengths and limitations of this study

► This will be the first systematic review to evaluate the diagnostic performance of artificial intelligence models using cross-sectional radiological images of the abdominopelvic cavity, identifying current limitations and evidence gaps, and thereby focusing future research efforts.
► Robust methodology will be undertaken including duplicate screening, data extraction and of risk of bias assessment.
► Findings may be limited by the inclusion of English language publications only.

## INTRODUCTION

In an era of 'Big Data', rapid developments in artificial intelligence (AI)-based technologies in medicine offer great potential to transform healthcare and improve patient outcomes.[1] The widespread adoption of digital data in healthcare has provided a vast amount of data to enable computer algorithms to extract relevant information and recognise complex patterns.[2] This includes quantitative (eg, laboratory values) and qualitative (eg, text-based electronic health records) data, as well as audio-visual data obtained from recordings from medical devices (eg, electrocardiograms, digital dictation). In a recent review, AI technologies were summarised as having an impact at three levels; clinicians, health systems and patients.[3] For clinicians, AI technologies can help interpret images more rapidly and accurately improving the performance of diagnostic, prognostic and management decisions.[4] For health systems, AI applications can improve workflow (eg, administrative jobs such as scheduling of operating rooms and clinic appointments). For patients, AI technologies can provide an opportunity for individuals to process their own data to promote health (eg, a smartwatch algorithm to detect a heart arrhythmia and the patient seeking appropriate medical attention).[3]

Medical imaging is considered a valuable source of diagnostic, prognostic and surveillance information. It also provides a pivotal role in supporting clinicians to perform procedural tasks. Images, however, have

traditionally been dependent on human interpretation and there is an increasingly limited number of interpreters.[5] There has been a surge of research exploring how AI technologies can be applied to medical images to support clinicians and provide greater efficacy and efficiency in clinical care.[6] One of the most promising clinical applications of AI has been in diagnostic imaging,[7] particularly for radiological[7–14] and endoscopic[15 16] investigations. AI diagnostic models have been used to detect pulmonary nodules,[12] liver lesions,[8] pancreatic cancer,[9] colorectal cancer[15] and hip fractures.[13 14] These advancements can be stratified by imaging modality (eg, ultrasound, radiography, CT, MRI) and anatomical region (eg, head and neck, thorax, abdomen and pelvis, upper and lower limbs),[17] organ[7] or specialty. However, most AI studies are currently a proof-of-concept, rather than a model deployed in the clinical setting to explore the potential benefit. Few prospective studies and randomised controlled trials (RCTs) evaluating the application of AI have been undertaken, and those which exist are at high risk of bias.[18]

Evidence from some clinical specialties, including neurosurgery[19] and gastroenterology[20] offer insight into some of the promises and pitfalls for AI technologies in healthcare. Pitfalls include AI algorithms that can be difficult or impossible to interpret (referred to as 'black box' techniques) and requiring large amounts of high-quality data which can be difficult to access, especially across institutions.[19] A review of the diagnostic accuracy of AI in radiological imaging of the abdominopelvic cavity is lacking. This could benefit a variety of different surgical specialties which employ diagnostic imaging for the abdominopelvic cavity. This systematic review aims to summarise the current research and critically appraise the diagnostic performance of AI models to diagnose disease from cross-sectional radiological images of the abdominopelvic cavity which may warrant an 'invasive procedure'[21] for 'therapeutic intent'. This will be compared with standard practice. The quality of this research will also be assessed, to identify current limitations and inform future research efforts.

## METHODS

### Protocol and registration

This protocol has been developed in accordance with the Preferred Reporting Items for Systematic Reviews and Meta-analyses (PRISMA) Protocols ('online supplemental file 1') guidelines.[22]

### Eligibility criteria

Primary research studies will be considered for eligibility using the PIRT framework (participants, index test(s), reference standard and target condition).[23]

### Participants

Adults with abdominopelvic cavity pathology diagnosed from cross-sectional radiological imaging confined to CT,

MRI and positron emission tomography (PET). Studies reporting endoscopy as an imaging modality will not be included in this review, as several reviews have already explored the performance of AI in this area.[16 24]

### Index test

Studies considering AI models as an intervention with the aim to provide a diagnosis.

### Reference standard

Standard practice.

### Target condition

Abdominopelvic cavity pathology which has had, or may warrant, an 'invasive procedure'[21] for 'therapeutic intent'.

### Exclusion criteria for the studies as follows

1. Secondary research studies (eg, editorials and systematic reviews), case reports and case series.
2. Absence of full text publications (eg, conference abstracts).
3. Non-English articles.
4. Animal studies.

### Outcome

The primary outcome is to evaluate the diagnostic performance of AI models using cross-sectional radiological images of the abdominopelvic cavity. The diagnostic performance will be referred to as previously defined: 'the ability of a test to discriminate between the target condition and health'.[25] Diagnostic measures of accuracy will include reported sensitivity, specificity and the area under the receiver operating characteristic curve.

### Information sources

An electronic search of OVID SP versions of Medline, EMBASE and the Cochrane Central Register of Controlled Trials will identify all potentially relevant studies published since 1 January 2012, using a predefined search strategy (online supplemental appendix S1). The cut-off from 1 January 2012 is to accommodate for the advancement in machine learning performance with the development of deep learning approaches, an approach previously adopted in the literature.[5]

### Search strategy and study selection

The search syntax will be developed with guidance from an information specialist using free text and Medical Subject Headings (MeSH) related to 'artificial intelligence', 'diagnostic imaging' and the 'abdominopelvic cavity' (online supplemental appendix S1). Database search results will be imported into the EndNote reference management software and duplicates will be removed.

Assessment of study eligibility will be a two-stage process. First, titles and abstracts will be screened for inclusion by two independent reviewers. Any identified conflicts will be resolved through discussion, including with the wider study team if required. Final eligibility will be assessed by full-text review of potentially eligible studies by the same

process. The screening will be facilitated by Rayyan software.[26] Reference lists of included studies will also be assessed for study eligibility.

## Data extraction and management

Eligible studies will undergo data extraction by two independent reviewers using a predesigned standardised proforma and data management software (REDCap V.9.5.23). A standardised form will be used, which will include the following categories:

1. Study characteristics: first author, journal, year of publication, country of origin, study design (eg, case control, RCTs) reporting of ethical approval, regulatory approval (eg, Medicines and Healthcare products Regulatory Agency) and patient and public involvement (PPI).
2. Patient characteristics: pathology studied and surgical specialty of pathology.
3. Input features: modality of radiological imaging (eg, CT, MRI and PET), AI model used, size of training model, comparator group used and size of this data set.
4. Outcomes: diagnostic measures of accuracy and method of validation.

## Risk of bias

Risk of bias of the primary diagnostic accuracy studies will be assessed using the QUADAS-2 tool.[27] This will be done independently by at least two authors of the study and disagreements resolved by the study team.

## Data synthesis

Search results and study selection will be presented in accordance with the PRISMA guidelines.[28] Due to the broad nature of the PIRT and anticipated high levels of heterogeneity for the primary outcome a meta-analysis of data is not planned. Findings will be presented as a descriptive summary and narrative synthesis and will be reported according to the Synthesis Without Meta-analysis guidelines.[29] The narrative synthesis will focus on the primary outcome with studies grouped by the modality of cross-sectional radiological imaging, pathology studied and surgical subspecialty.

## Patient and public involvement

As part of the wider programme of work (Bristol Biomedical Research Centre, National Institute for Health Research (NIHR) Bristol BRC), patients and the public were consulted on their views of AI being used to guide doctors to make decisions about treatment. PPI will be sought for the dissemination of this systematic review.

## Ethics and dissemination

Ethical approval is not required for the systematic review, as no primary data is collected. The review will be disseminated at relevant conferences, on social media and published in a peer-review journal.

**Correction notice** This article has been corrected since it first published. Funding statement has been updated.

**Acknowledgements** The authors would like to thank Ms Catherine Borwick, an Information Specialist from the University of Bristol, for her expertise in developing the search strategy. The authors are also grateful to Dr Christin Hoffmann for leading patient and public feedback sessions as part of the wider programme of this research.

**Contributors** GEF and NSB conceived the idea for this systematic review. The search strategy was developed by all authors. GEF drafted the manuscript protocol (guarantor of review), and it was critically appraised and revised by NSB, CH, MPC, NJS and RCM. All authors approved the final manuscript before submission.

**Funding** NSB is funded by an MRC Clinical Scientist Award (MR/S001751/1). This study was also supported by the NIHR Biomedical Research Centre at University Hospitals Bristol and Weston NHS Foundation Trust and the University of Bristol. The views expressed are those of the author(s) and not necessarily those of the NIHR or the Department of Health and Social Care.

**Competing interests** None declared.

**Patient consent for publication** Not applicable.

**Provenance and peer review** Not commissioned; externally peer reviewed.

**ORCID iDs**
George E Fowler http://orcid.org/0000-0002-4133-802X
Rhiannon C Macefield http://orcid.org/0000-0002-6606-5427
Natalie S Blencowe http://orcid.org/0000-0002-6111-2175

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
