## [Reviewer comments · BMJ Open]

ARTICLE DETAILS

TITLE (PROVISIONAL)	Artificial intelligence as a diagnostic aid in cross-sectional radiological imaging of the abdominopelvic cavity: A protocol for a systematic review
AUTHORS	Fowler, George; Macefield, Rhiannon; Hardacre, Conor; Callaway, Mark; Smart, Neil; Blencowe, Natalie

VERSION 1 – REVIEW

REVIEWER	Perrone, Myriam University of Bologna, Division of Oncologic Gynecology, IRCCS-Azienda Ospedaliero-Universitaria di Bologna
REVIEW RETURNED	19-Jun-2021

GENERAL COMMENTS	Clear and concise text all the parameters for proper review have been met. Good luck with your work.
--

REVIEWER	Lim, Gilbert National University of Singapore, School of Computing
REVIEW RETURNED	19-Jun-2021

GENERAL COMMENTS	This manuscript describes the protocol for a descriptive/narrative systematic review, on artificial intelligence (AI) for cross-sectional imaging of the abdominopelvic cavity. A few comments might be considered: 1. As it is stated that "studies reporting endoscopy as an imaging modality will not be included in this review" (Page 12), and that the review will be focused on radiological imaging, it might be considered to specify radiological imaging in the article title.2. The search strategy as detailed in Appendix S1 might be referenced in the main text (Page 14) where appropriate.3. Related to the above, it might be considered to include broader databases (e.g. Scopus, Web of Science, etc.), and to consider the reference list from articles selected from the initial primary search, which appears common practice for such systematic reviews.4. It is stated that "Eligible studies will undergo data extraction using a pre-designed standardized proforma and data management software (REDCap version 9.5.23). This will be done independently by at least two reviewers for at least 10% of the articles and then verified by a second thereafter" (Page 14); the significance of having at least two reviewers for at least 10% of the
---

	articles, might be explained further. Would this be part of some quality assurance methodology?
--	---

REVIEWER	Dwivedi, Shweta Shukla Jabalpur Medical College
REVIEW RETURNED	26-Jul-2021

GENERAL COMMENTS	Manuscript needs major revision at every part. Methods are defined well , Inclusion and exclusion criteria , limitations are well marked . Manuscript is running out of track without its design of systematic review . So myself suggesting you , first look and make a research that how to write a systematic review . That's the main lacking overhere . I would like to ACKNOWLEDGE DR. ATUL DWIVEDI , who thoroughly provide me help to review this manuscript .
---

REVIEWER	Islam, Md. Mohaimenul Taipei Medical University
REVIEW RETURNED	28-Jul-2021

GENERAL COMMENTS	Thanks for giving me an opportunity to review this manuscript. It is well organized and presented clearly. However, I have some minor comments:  1. Author should provide result section what would be their primary outcome and will they present their outcome based on image modalities, namely PET, CT and MRI. 2. In the method section search terms are incomplete "Medical Subject Headings (MeSH) related to "artificial intelligence", "diagnostic imaging" and the "abdominopelvic cavity". Authors also provided supplementary to show how they will search by various algorithms but most of them are useless. For example, we can not use random forest for image analysis. We can only use CNN and support sector machine model for image analysis. Don't need to search by other classification algorithms.
---

VERSION 1 – AUTHOR RESPONSE

Reviewer #1:

Clear and concise text all the parameters for proper review have been met. Good luck with your work.

Response: We thank the reviewer for their support.

Reviewer #2:

1. As it is stated that "studies reporting endoscopy as an imaging modality will not be included in this review" (Page 12), and that the review will be focused on radiological imaging, it might be considered to specify radiological imaging in the article title.

Response: Thank you for this suggestion. We have revised the title accordingly.

2. The search strategy as detailed in Appendix S1 might be referenced in the main text (Page 14) where appropriate.

Response: Thank you for this suggestion. Appendix S1 is now also referenced within the subheading titled "Search strategy and study selection".

3. Related to the above, it might be considered to include broader databases (e.g. Scopus, Web of Science, etc.), and to consider the reference list from articles selected from the initial primary search, which appears common practice for such systematic reviews.

Response: We are grateful for these suggestions. We had considered other databases and chose the three listed databases (Medline, EMBASE and the Cochrane Central Register of Controlled Trials (CENTRAL)) with professional guidance from our medical librarian to facilitate a broad search for this systematic review. We have incorporated your suggestion to include screening the reference lists from included articles.

4. It is stated that "Eligible studies will undergo data extraction using a pre-designed standardized proforma and data management software (REDCap version 9.5.23). This will be done independently by at least two reviewers for at least 10% of the articles and then verified by a second thereafter" (Page 14); the significance of having at least two reviewers for at least 10% of the articles, might be explained further. Would this be part of some quality assurance methodology?

Response: We agree with the reviewer that this would add strength to the systematic review methodology. We have now amended the protocol to perform data extraction by at least two people for all articles, rather than 10%. This is supported by the Cochrane handbook (<https://training.cochrane.org/handbook/current/chapter-05#section-5-5>) for data that is critical to the interpretation of the results, which includes outcome data on diagnostic accuracy.

Reviewer #4:

Thanks for giving me an opportunity to review this manuscript. It is well organized and presented clearly. However, I have some minor comments:

1. Author should provide result section what would be their primary outcome and will they present their outcome based on image modalities, namely PET, CT and MRI.

Response: Thank you for your comments and in accordance with the PRISMA-P and PRIMSA guidelines, we will provide the results section with the systematic review, rather than in the protocol, which is also in keeping with other recently published systematic review protocols. The results section will focus on the primary outcome (diagnostic performance of the AI models), with studies grouped by the modality of cross-sectional radiological imaging (i.e. CT, MRI or PET), pathology studied (i.e. benign, malignant etc) and surgical subspecialty, as already outlined in the protocol.

2. In the method section search terms are incomplete "Medical Subject Headings (MeSH) related to "artificial intelligence", "diagnostic imaging" and the "abdominopelvic cavity". Authors also provided supplementary to show how they will search by various algorithms but most of them are useless. For example, we can not use random forest for image analysis. We can only use CNN and support vector machine model for image analysis. Don't need to search by other classification algorithms.

Response: We are grateful for this comment and acknowledge are search strategy could be less

detailed and subsequently less labour intensive for the study selection process. However, this could be at the detriment of missing a few papers which meet the inclusion criteria. This would include a study that has used a random forest classification framework in combination with histogram analysis to detect abnormalities of the adrenal glands (Saiprasad G, Chang CI, Safdar N, Saenz N, Siegel E. Adrenal gland abnormality detection using random forest classification. J Digit Imaging. 2013 Oct;26(5):891-7. PMID: 23344259).

We hope that you will find the revised manuscript appropriate for publication in the BMJ Open.

VERSION 2 – REVIEW

REVIEWER	Lim, Gilbert National University of Singapore, School of Computing
REVIEW RETURNED	28-Aug-2021
GENERAL COMMENTS	We thank the authors for considering our suggestions, and have no further concerns.
REVIEWER	Islam, Md. Mohaimenul Taipei Medical University
REVIEW RETURNED	29-Aug-2021
GENERAL COMMENTS	Thanks for revised version. Authors have answered all of my questions. Current version can be considered for publication.